# Effect of Acetylation on Physicochemical and Functional Properties of Commercial Pumpkin Protein Concentrate

**DOI:** 10.3390/molecules26061575

**Published:** 2021-03-12

**Authors:** Joanna Miedzianka, Aleksandra Zambrowicz, Magdalena Zielińska-Dawidziak, Wioletta Drożdż, Agnieszka Nemś

**Affiliations:** 1Department of Food Storage and Technology, Wroclaw University of Environmental and Life Sciences, 37 Chelmonskiego Street, 51-630 Wrocław, Poland; wioletta.drozdz@upwr.edu.pl (W.D.); agnieszka.nems@upwr.edu.pl (A.N.); 2Department of Functional Products Development, Wroclaw University of Environmental and Life Sciences, 37 Chelmonskiego Street, 51-630 Wrocław, Poland; aleksandra.zambrowicz@upwr.edu.pl; 3Department of Biochemistry and Food Analysis, Poznań University of Life Sciences, 48 Mazowiecka Street, 60-623 Poznań, Poland; magdalena.zielinska-dawidziak@up.poznan.pl

**Keywords:** pumpkin protein concentrates, acetylation, chemical composition, amino acid profile, in vitro digestibility, SDS-PAGE, functional properties

## Abstract

The purpose of the present study was to determine the effects of acetylation with different doses of acetic anhydride on the chemical composition and chosen functional properties of commercial pumpkin protein concentrate (PPC). The total protein content decreased as compared to unmodified samples. Electrophoretic analysis revealed that in the acetylated pumpkin protein, the content of the heaviest protein (35 kDa) decreased in line with increasing concentrations of modifying reagent. Acetylation of PPC caused a significant increase in water-binding and oil-absorption capacity and for emulsifying properties even at the dose of 0.4 mL/g. Additionally, an increase in foaming capacity was demonstrated for preparations obtained with 2.0 mL/g of acetic anhydride, whereas acetylation with 0.4 and 1.0 mL/g caused a decrease in protein solubility as compared to native PPC.

## 1. Introduction

Pumpkin seeds are a source of fat (37–45%) and protein (25–37%), which is distinguished by a high content of indispensable amino acids. They also contain about 11% carbohydrates and 6% dietary fibre, as well as many vitamins and minerals [1]. In Europe, pumpkin (*Cucurbita pepo* L.) is cultivated in the middle south region, primarily for oil production from seed, where the main by-product from oil extraction is defatted pumpkin cake, which contains up to 65% protein [2,3]. Due to the high protein content, this by-product left over from pumpkin oil seed extraction is considered by many authors as one of the most attractive and promising sources of vegetable proteins with proven health-promoting and functional effects [4,5].

One of the uses of pumpkin cake, apart from enrichment with amino acids, is protein preparations constituting newly developed functional ingredients that enrich the nutritional properties of food products. Favourable results have been obtained for extraction of functional proteins from pumpkin seed cakes under alkaline conditions. In this technique, pumpkin cakes are treated with an NaOH solution at pH 8–10, after which the solution is centrifuged and the proteins are precipitated therefrom at the isoelectric point. In this way, isolates containing as much as 80–90% protein can be obtained with high yield. Additionally, this method is used to obtain the broad spectrum of proteins found in pumpkin seeds [5].

Widely available pumpkin protein concentrates and isolates are highly appreciated, despite their poor functional properties. During the manufacture of pumpkin protein preparations, the applied heat treatments cause extensive protein denaturation and the resulting preparations are characterized mainly by poor solubility, limiting their incorporation into food systems. Therefore, a roadblock to the large-scale use of plant proteins is their poor solubility under mildly acidic (pH 3–6) conditions. This excludes their use in acidic foods such as coffee whiteners, acidic beverages, yogurts and pourable and non-pourable dressings. The weak solubility of proteins, however, sets limits for their utilization in formulated food systems and so the solubilization of pumpkin protein preparations has been attempted to extend their usefulness in the food industry [6].

One of the most convenient and frequently used methods for altering the functional properties of proteins is acetylation. This chemical modification of proteins involves the reaction of acetic anhydride with the ε-amino group of lysine and other nucleophilic groups such as phenolic (e.g., tyrosine) and aliphatic hydroxyl groups (e.g., serine and threonine) [7]. The acetic anhydride (CAS number 108-24-7) used here is a chemical compound widely used in organic synthesis. It is a colourless liquid that smells strongly of acetic acid. Acetylation of proteins in the range of 0.2–1.0 mL/g protein has been widely known for more than 40 years [8,9,10,11]; modified proteins have been applied in the preparation of products such as coffee whiteners [12], flavouring agents for roasted meat [13] and carbonated beverages [14]. Additionally, recently, studies on improving functionality by acylation have been applied to many food proteins including spray-dried egg white [15], oat protein isolates [16] and fish myofibrillar protein [17]. However, in the literature, there is no report available on the functionality of acetylated commercial pumpkin protein concentrate. This chemical modification is considered safe because acetic anhydride is rapidly hydrolysed (half-life 4.4 min) to acetic acid, which is readily biodegradable. In the atmosphere, it is converted to acetic acid, which is subject to photooxidative degradation (half-life 22 days). Toxicity to aquatic organisms is moderate (18 to 3400 mg/L), but it persists only for a short time due to its rapid hydrolysis to acetate/acetic acid. Acetic acid is further used as a food and animal feed additive or a preservative in pickles [18].

For the above-stated reason, this investigation through chemical modification has been undertaken in order to improve the functional properties of pumpkin protein preparations, mainly water-binding, oil-absorption capacities, solubility, foaming and emulsifying properties, while limiting the negative impact on changes in the chemical composition and digestibility of proteins. Therefore, the purpose of the present study was to determine the effect of the anhydride-to-protein ratio of acetylation on the chemical composition and functional properties of commercial pumpkin protein preparations. It is hoped that the data will provide information on acetylated derivatives, which could be useful in experiments concerning the chemical modification of commercial pumpkin protein preparations appropriated for food purposes.

## 2. Results and Discussion

### 2.1. Chemical Composition

The chemical composition of native and acetylated pumpkin protein concentrate (PPC) is presented in Table 1. All analysed samples differed in dry matter, fat and ash content. The dry matter content of PPC preparations ranged from 94.90% to 99.01%. This is in line with the results obtained by Rutkowski and Kozłowska [19], who reported a water content for protein preparations not exceeding 10%. In this study, the application of 1.0 and 2.0 mL/g acetic anhydride had a significant impact on the dry matter content of PPC. The protein content in the analysed commercial pumpkin preparation reached a value of 65.11%. Based on the data found in the literature [20], it can be concluded that these proteins were extracted and purified in a neutral or acidic environment. Moreover, according to Ozuna and León-Galván [5], the composition of protein preparations depends on the chemical composition and characteristics of the proteins contained in the raw material. In most cases, acetylation had no effect on the protein content. No statistically significant differences were found between the control and the most experimental samples. Along with an increase in acetic anhydride concentration from 0.0 to 2.0 mL/g, there was no observed tendency to a decrease in the total protein share in the analysed preparations, except for the sample modified with 1.0 mL/g acetic anhydride, where a statistically significant (*p* < 0.05) decrease was noted. Under these conditions, acetylation caused a decrease in total protein content in modified samples, from 63.24 to 53.52 g/100 g. These results are in line with those of other authors who acetylated protein preparations, such as El-Adawy [21], Lawal and Adebowale [22], Lawal et al. [23] and Miedzianka et al. [24]. However, our results disagree with those of Khader et al. [25] and Zedan et al. [26] in which the total protein percentage increased after acetylation. The different research results indicate that the effect of acetylation on protein content is not unequivocal.

Acetylation reduced the fat content in modified PPC. As shown in Table 1, the fat content ranged from an average of 9.17 g/100 g before chemical modification to an average of 8.89 g/100 g in modified samples. However, the acetylated samples were not significantly differed in fat content. Increasing the amount of anhydride did not affect the level of fat.

The total ash content in native PPC (8.36 g/100 g) was similar to the 9.09 g/100 g found by Zduńczyk et al. [27] in pumpkin oil seed cake. The results are in line with statements that plant-derived protein preparations usually have an ash content of less than 10%, because they contain relatively fewer minerals than preparations of animal origin, and their availability is additionally limited by the presence of chelating compounds. Moreover, the total ash content in protein preparations is mainly influenced by the quality of the raw material, which is largely due to the plant variety, the degree of purification and insulation conditions (pressing technology or the type of press used). The ash content in acetylated samples differed significantly (Table 1). A statistically different decreasing trend was found on the ash content for higher anhydride-to-protein ratios. It ranged from 5.49 g/100 g (in samples modified with 2.0 mL/g) to 8.68 g/100 g (in samples acetylated with 0.4 mL/g). Similar findings were observed by Lawal and Adebowale [22], who acetylated the jack bean protein. This decreasing ash content in acetylated protein preparations can be associated with more frequent removal of an excess modifying reagent (samples modified in the amount of 2.0 mL/g of protein were washed 5 times).

### 2.2. Amino Acid Composition

One of the reasons proteins exhibit such different functional properties is that their amino acid composition is varied [28]. This also influences the functional properties of a protein depending on how they are arranged in the polypeptide chain. As presented in Table 2, the dominant amino acids found in PPC were aspartic acid (7.04 g/100 g protein), glutamic acid (21.03 g/100 g protein), arginine (16.69 g/100 g protein) and leucine (6.67 g/100 g protein). Similar levels of amino acids in protein preparations obtained from pumpkin oilseed cake were reported by Ozuna and León-Galván [5]. Of the indispensable amino acids, lysine (3.19 g/100 g protein) and sulphur amino acids (2.04 g/100 g protein) were the least noted in PPC. These results provide support for Kotecka-Majchrzak et al.’s [29] theory that the amino acids present at the highest concentrations in oilseed proteins are leucine and valine, whereas sulphur-containing amino acids and hydrophobic tryptophan are present in the lowest amounts. Moreover, the amino acid profile depends on the methods used in both the extraction and coagulation of protein preparations [30]. Acetylation of PPC decreased all amino acids (except lysine and alanine) in preparations. The sum of amino acids decreased from 88.63 to 79.88 g/100 g protein after modification with 2.0 mL/g. A statistically significant decrease was observed in all analysed amino acids, particularly in lysine, tyrosine, methionine with cysteine and phenylalanine with threonine. These results confirm the data presented by other authors [11,24,31].

### 2.3. Degree of Acetylation and In Vitro Digestibility

As shown in Table 3, along with an increase in acetic anhydride concentration from 0.0 to 2.0 mL/g, there was a higher degree of N-acetylation in analysed preparations. This phenomenon was due to the number of ε-amino groups involved in the acetylation reaction being reduced severely with the increase in the anhydride level, leading to an increase in the degree of N-acetylation. Moreover, this increase might be correlated with process conditions and the source of protein subjected to chemical modification [32]. Additionally, Achouri and Zhang [33] stated that the extent of amino group acetylation in a polypeptide mixture depends markedly on the amount of acetic anhydride used. However, 78.67% of the ε-amino groups was acetylated with 2.0 mL/g. This means that a complete blockage of the amino acid residues is possible only in the presence of a high dose of acetic anhydride.

After digestion of the analysed preparations, protein nitrogen extraction ranged from 21.4% (for samples acetylated with dose 0.4 mL/g) only up to 25.31% (for samples acetylated with dose 1.0 mL/g) (Table 3). The amount of protein nitrogen determined in the supernatant obtained after the two-stage digestion contained nitrogen compounds and not those extracted from the sample proteins. Thus, the determination should reflect the presence of short peptides, possible to the absorption by the intestine enterocytes. The digestion step did not include digestion by the enzymes excreted by enterocytes. The cleavage specificity of porcine pepsin includes peptides with an aromatic amino acid, preferentially at carboxylic groups of the amino acid, especially if the other residue is also an aromatic or a dicarboxylic amino acid and does not destroy bonds containing valine, alanine or glycine linkages [34], while trypsin cleaves the peptide chain on the C-terminal side of lysine and arginine amino acid residues. Similar findings were reported by Shukla [35]. However, the applied method is definitely more precise than the method proposed by Salgó et al. [36] because that determines only peptides, not protein, extracted from the studied material. Thus, the results obtained should not be compared with those obtained for pH changes after placement of the sample in the intestine solutions. We showed that the use of 0.4 mL/g acetic anhydride significantly reduces the digestibility of modified PPC in comparison to native PPC, while acetylation with 1.0 mL/g acetic anhydride significantly improves the digestibility of PPC. Therefore, the effect of acetylation on the digestibility of PPC preparations cannot be clearly determined. However, according to Bergner et al. [37], acetylation of proteins does not reduce either apparent or true N digestibility. Additionally, this modification of food proteins does not influence the nutritive value.

### 2.4. SDS-PAGE

The resulting native and acetylated PPC were analysed by SDS-PAGE (Figure 1). Electrophoretic analyses revealed that the PPC contained a predominant protein of about 35 kDa·MW. As we can see on the electropherogram, proteins of about 18–22 kDa·MW accompanied this main protein. According to previous research studies, cucurbitin, the major storage protein in pumpkin seed, is a hexamer with a molecular size of about 325 kDa, consisting of two major bands (MW d 22 kDa) corresponding to acidic and basic polypeptides [38,39,40,41]. The vast majority of cucurbitins are salt-soluble globulins with a sedimentation index of about 11–12S [42]. They are accompanied by albumin with a sedimentation index of 2S and a mass of about 12.5 kDa, composed of two subunits: a smaller one, about 4.8 kDa and above, and another with a mass of 7.9 kDa [43,44]. Moreover, cucurbitins have proven antimicrobial, antitumour and anti-inflammatory effects [45]. Analysis revealed that in the acetylated pumpkin protein the content of the heaviest protein (35 kDa) decreased in line with increasing concentrations of modifying reagent. The addition of 0.4–2.0 mL acetic anhydride per g of protein resulted in removal of a protein of about 18 kDa·MW and, at the same time, resulted in an increase in the participation of a protein of about 20 kDa·MW in the composition of acetylated pumpkin protein preparations. Additionally, the proteins of about 35 W and 18 kDa·MW were more susceptible than the protein of about 22 kDa·MW to the presence of the modifying reagent. Furthermore, the intensity of the protein patterns decreased in line with increasing concentrations of modifying reagent, which confirms the reduction in protein content in acetylated pumpkin protein preparations. These results are not in line with those of Zhao et al. [16], who found that acetylation of oat protein isolate with different anhydride-to-protein ratios left the molecular weight of the proteins almost unchanged, as compared to native protein isolate.

### 2.5. Functional Properties

#### 2.5.1. Protein Solubility Index

Plant protein preparations used in food production should be characterized not only by a favourable chemical composition, but also by optimal functional properties, of which the most significant for other functional applications of the proteins is solubility (PS). PS depends on the pH value and ionic strength as well as the method of obtaining protein preparations. As shown in Figure 2, the applied chemical modification of the protein caused statistically significant (*p* < 0.05) changes in the PS of analysed preparations. The solubility profile of native PPC indicates that PS reduced as the pH increased from 2 to 4 (from 10.16% to 6.48%), which corresponds to its isoelectric point, after which an increase in pH increased PS progressively (to 62.55%). This solubility profile of pumpkin proteins is U-shaped, common for oil seeds [40,41]. Other authors [46,47,48,49] have emphasized that the solubility of the proteins is strongly influenced by pH and ionic strength, with higher values in the alkaline pH regions, and by the type of solvent used to degrease the seeds during preparation of the raw material. Increased PS was observed only for the preparation modified with 2.0 mL/g. This preparation had the highest solubility of proteins at pH = 12 (49.42%) and the lowest at pH = 4 (20.11%). This phenomenon can be explained by higher protein degradation in this sample and by the increased interaction between proteins and water, decreasing protein–protein interactions and, hence, increasing their solubility [48]. These results are in line with the studies presented by Lawal and Adebowale [22] and El-Adawy [21] who analysed jack bean and acylated mung bean protein isolate, respectively. These well-soluble proteins could be used in the production of meat products, improving the stability of the emulsion [19]. Furthermore, acetylation with 0.4 and 1.0 mL/g caused a decrease in PS, as compared to native PPC.

#### 2.5.2. Water-Binding Capacity and Oil-Absorption Capacity

The water-binding capacity (WBC) of native and acetylated PPC is shown in Figure 3. Acetylation significantly increased (*p* ≤ 0.05) the WBC at all levels of modification compared to native preparation (2.54 g/g). Similar findings were present by El-Adawy and Taha [49] who obtained water absorption of pumpkin flour at the level of 2.51 g/g of preparation. According to Rodriques et al. [50], pumpkin protein preparations show a low WBC, which may be due to the relatively high fat content and low availability of non-polar amino acids (i.e., glycine, proline, valine and tyrosine). Depending on the anhydride-to-protein ratio, the WBC of the modified concentrates was in the range 4.89–5.62 g/g. However, significantly the highest WBC was noted for sample modified with 0.4 mL/g of acetic anhydride. This resulted from sufficient exposure of hydrophilic groups and the higher amounts of polar amino acids present in protein particles. This phenomenon is confirmed in the decreased PS of this sample, since poorly soluble proteins have been found to exhibit good water absorption [6]. Acetylation can cause dissociation and unfolding of the protein; therefore, protein–protein reactions are limited, and protein-water interactions are facilitated [22]. Similar findings presenting an increase in WBC by acetylation have been reported by El-Adawy [21] for mung bean, by Liu and Hung [51] for chickpea proteins, by Dua et al. [52] for rapeseed, by Bora [53] for lentil and by Miedzianka et al. [24] for potato. Due to its high WBC, acetylated PPC could be used in viscous products (dough, processed cheese) [19].

As shown in Figure 3, acetylated pumpkin protein preparations were characterized by a much higher oil-absorption capacity (OAC), in the range 4.7–4.94 mL/g, as compared to native PPC not subjected to the modification process (0.97 mL/g), independent of the anhydride-to-protein ratio. Therefore, it can be assumed that native pumpkin protein is characterized by high density and particle size, which is why it absorbs less oil [48,54]. The results revealed that even at a dose of 0.4 mL/g, the OAC of analysed protein concentrates increased fourfold. This fact may be related to the introduction of lipophilic groups in the pumpkin protein molecules during acetylation [55]. Generally, the OAC of acetylated protein preparations may be attributed to the degree of denaturation due to the method of obtaining them as well as on the type of protein sample used for modification [56]. Similar findings were stated by Lawal and Adebowale [22] and by Miedzianka et al. [24]. The obtained acetylated preparations characterized by high OAC may, therefore, be applied as meat product extenders.

#### 2.5.3. Foaming Properties

The effect of anhydride-to-protein ratio on the foaming capacity (FC) at different pH of native and acetylated PPC is shown in Figure 4. These data reveal that the commercially purchased preparation is characterized by very low FC, regardless of pH, which may be due to the compact, globular structure and high molecular mass of these proteins, which inhibits their ability to reorient effectively. These results are not in line with those of El-Adawy and Taha [49], who found pumpkin flour was characterized by a significantly higher capacity to create foam (18.65%). Additionally, Rezig et al. [46] stated that, in fact, the mechanisms causing the formation and stability of foam in the presence of single pumpkin protein molecules and protein aggregates are not yet fully understood and insufficiently described in the literature. Acetylation with higher doses of acetic anhydride (1.0 and 2.0 mL/g) significantly increased (*p* ≤ 0.05) the FC, mainly at pH = 2, pH = 8 and pH = 12. This phenomenon is related to an increase in the net charge of the protein molecules, which weakens hydrophobic interactions and increases protein flexibility [26]. Similar findings were stated by El-Adawy [21] for acetylated mung bean. Foam was not noticeable at pH = 4 and pH = 6 in native and acetylated PPC, which was related to their low solubility at different pH.

Table 4 presents the effect of anhydride-to-protein ratio on the foam stability (FS) of native and acetylated PPC. Generally, commercial pumpkin protein preparation had weak FS, and the formed foam was stable only at pH = 2, pH = 10 and pH = 12. For foam to be stable, a thick, flexible, continuous and air-permeable protein film is required around each air bubble [46]. Additionally, Wani et al. [57] analysed the FC of albumin, globulin, glutelin and prolamin in two varieties of watermelon belonging to the same botanical family as the pumpkin (Cucurbitaceae). They showed that the FS in both varieties is the highest for the albumin fraction. The differences in stability were attributed to differences in the protein structure. Acetylation with different doses of acetic anhydride had a negative effect on the FS of the analysed samples. After acetylation, the FS was decreased probably due to the negative charges imparted during modification, causing the protein molecule to unfold [6,21,23]. However, stabilization of the formed foam increased only at pH = 12. In general, preparations characterized by low molecular weight, high surface hydrophobicity, good solubility and greater denaturability exhibit improved foaming properties [58].

#### 2.5.4. Emulsifying Properties

The effect of anhydride-to-protein ratio on the emulsifying activity (EA) and stability (ES) at different pH levels of native and acetylated PPC is shown in Figure 5 and Figure 6, respectively. Commercially purchased pumpkin protein preparation was characterised by poor EA and ES, mainly at a pH ranging between 4 and 10 (on average, it valued less than 5%). This confirmed the fact that highly insoluble proteins are not good emulsifiers [59]. The better EA for PPC was found at extreme pH, i.e., 2 and 12, where it was noted as 15.62 and 30.61%, respectively. The formed emulsion with PPC retained its properties after heating to 80 °C. Acetylation improved the EA, especially when using the acetic anhydride dose of 0.4 and 2.0 mL/g, in the pH ranging between 6 and 10. One possible explanation for this is the increase in solubility at dose of 2.0 mL acetic anhydride/g, where these changes were found to facilitate the diffusion of protein at oil–water interface [60]. Chemical modification with 1.0 mL/g caused EA similar to PPC, independently from pH range. Similar observations for acetylation the plant-derived proteins were presented by Bora [53] and Miedzianka et al. [24]. In contrast, El-Adawy [21] observed lower EA of preparations obtained as an effect of acetylation of mung bean protein with doses of acetic anhydride higher than 0.6 g/g protein. Furthermore, El-Adawy [21] proved that the best effect for mung bean proteins is obtained using 0.4–0.6 g/g of acetic anhydride. This proves that apart from the conditions of acetylation, the origin (properties) of proteins influences the change in their functional properties.

## 3. Materials and Methods

### 3.1. Materials and Chemicals

The material analysed was commercial plant-derived protein concentrate from pumpkin seeds (PPC), purchased from the Diet Food company (Opatówek, Poland). Acetic anhydride, 2,4,6-trinitrobenzenesulfonic acid (TNBS), pepsin (60,000 U), bile salts and pancreatin for porcine pancreas were obtained from Sigma-Aldrich (St. Louis, MO, USA). All chemicals used in the experiment were of analytical grade.

### 3.2. Acetylation of PPC

Acetylated PPC samples were prepared using the method of Miedzianka et al. [24] with slight modifications. The pumpkin protein preparation was acetylated with acetic anhydride by adding different concentrations of modifying reagent (0.4, 1.0 and 2.0 mL of anhydride per 1 g of protein contained in the preparation) to the 1% aqueous suspension. The reaction took place over 30–90 min, and the pH was established at pH 7.5–8 by dropwise addition of 1 M NaOH. The solution was constantly monitored for changes in pH and maintained at the assumed pH with constant stirring. After this time, the precipitate was separated from the supernatant by centrifugation using a Biofuge 28 RS centrifuge (5260 rpm/min for 15 min, Heraeus Sepatech, Osterode, Germany). Excess modifying reagent was removed from the resulting precipitate, i.e., the precipitate was mixed with water until the liquid’s conductance was close to that of distilled water (3–5 times). Then, it was bathed with distilled water until the conductance of liquid was similar to that of distilled water. Then, the protein was freeze-dried at a pressure of 63 Pa, with the heating shelves at 50 °C, for 24 h using a Christ Alpha 1-4 LSCplus (Osterode am Hatz, Germany), sieved through a sieve with a pore size of 420 μm and stored in a sealed plastic container at about −20 °C until further analysis. Untreated native PPC was used as the control. Acetylation was performed in two technological repetitions.

### 3.3. Basic Chemical Composition

In order to determine the moisture content, approximately 2 g of a sample was placed into a pre-weighed vessel. Samples were dried at 105 °C until a constant weight was reached [61]. The total protein content was evaluated according to the Kjeldahl method of the Association of Analytical Chemists [62]. Approximately 0.5 g of material was hydrolysed with 25 mL concentrated sulfuric acid (H_2_SO_4_) containing one catalyst tablet in a heat block (Büchi Digestion Unit K-424, Labortechnik AG, Flawil, Switzerland) at 370 °C for 2 h. After cooling, H_2_O was added to the hydrolysates before neutralization, using a Büchi Distillation Unit K-355 (Athens, Greece) and titration. The protein content was calculated by multiplying the percentage of nitrogen content by a factor of 6.25 [63]. Fat content was determined according to the standard method of the association of Official Analytical Chemists International [64]. A sample of 2 g of material was hydrolysed using 4 N HCl. Fat extraction and solvent (diethyl ether) removal were performed in an automated Soxhlet apparatus B-811 (Büchi Labortechnik AG, Flawil, Switzerland); the extraction time was 180 min. Samples for determining the ash content were heated gradually to 550 °C, and the residues were weighed [62].

### 3.4. Amino Acid Composition

The amino acid composition of native and acetylated PPC was determined by ion-exchange chromatography after 23 h hydrolysis with 6 N HCl at 110 °C, according to the method described by Miedzianka et al. [24]. The hydrolysed amino acids were determined using an AAA-400 analyser (INGOS, Prague, Czech Republic). A photometric detector was used, working at two wavelengths, 440 and 570 nm. A column of 350 × 3.7 mm, packed with ion exchanger Ostion LG ANB (INGOS, Prague, Czech Republic), was utilized. Column temperature was kept at 60–74 °C and the detector at 121 °C. The prepared samples were analysed using the ninhydrine method. No analysis of tryptophan was carried out.

### 3.5. Measurement of Degree of N-Acylation

The measurement of degree of N-acylation was prepared, as described by Habeeb [65]. The procedure involved the addition of 1 cm^3^ of 4% NaHCO_3_ solution and 1 cm^3^ of 0.1% TNBS solution to protein suspensions. The samples were heated in a water bath at 60 °C for 2 h. Then, they were cooled down to room temperature. Next, 1 cm^3^ of 10% SDS and 0.5 cm^3^ of 0.1 N HCl were added to protein solutions. The absorbance of solutions was read at 335 nm in a Rayleigh UV-2601 PC spectrophotometer (Beijing, China) against a reagent blank. The absorbance of the control protein concentrate was set equal to 100% free amino groups, and the extent of acetylation of the modified samples was calculated based on the decrease in absorbance, because fewer amino groups were able to react with the TNBS reagent.

### 3.6. Digestibility of Protein Preparations

The digestibility of the native and acetylated PPC were determined with the method simulating multienzymatic two-stage digestion [66,67]. The oral digestion and large intestine steps were omitted as irrelevant to protein digestion. The digestion in the stomach was simulated by introducing the sample (~0.5 g) into the medium imitating the stomach acidic environment (by decreasing water pH down to 2.0 by 1 N HCl) and containing pepsin (60,000 U). Gastric digestion was carried out at 37 °C for 2 h. Then, the pH was increased up to 7.4 by 0.1 M NHCO_3_, and mixture was enriched in bile salts, acting as a surfactant (0.03 g) and porcine pancreatin (0.005 g), which contains proteases (trypsin, protease A, ribonuclease), amylase and lipase. The second digestion step was performed again at 37 °C for 2 h. The digested sample was centrifuged. The remaining proteins present in the supernatant—extracted and not digested—were precipitated with trichloroacetic acid. In this prepared supernatant, with the use of the Kjeldahl method, protein nitrogen was determined [68] and was related in percentage to the amount of protein nitrogen introduced with the sample into the digestion analysis.

### 3.7. Sodium Dodecyl Sulfate–Polyacrylamide Gel Electrophoresis (SDS-PAGE)

SDS-PAGE analysis was performed according to the method of Laemmli [69]. The protein samples were diluted (10 mg/mL) with the buffer containing SDS and β-mercaptoethanol as reducing reagents and denatured. Then, the samples were loaded (10 μL) onto gel slabs (12%). At the end of analysis, gel slabs were stained with Coomassie Brilliant G-250 dye. The molecular masses of analysed proteins were estimated from the response curve of the mobility of molecular weight standards versus logMW.

### 3.8. Determination of Functional Properties

#### 3.8.1. Effect of pH on Protein Solubility

The pH-solubility profile (PS) of native and acetylated PPC was determined according to the method of Achouri et al. [70] with slight modifications. Briefly, 750 mg of preparation was weighed in the tube, 15 mL of distilled water was added, and then, the pH was adjusted (2, 4, 6, 8, 10 or 12) using either 0.5 M NaOH or 0.5 M HCl. The protein solutions were shaken at room temperature for 30 min and successively centrifuged at 4.500 g for 15 min (Rotofix 32A by Hettich, Tuttlingen, Germany). The protein content of the supernatants was determined by Kjeldahl method. Protein solubility was calculated as (Equation (1)):(1)$$\mathrm{PS}=\frac{\mathrm{PCS}}{\mathrm{TPC}}\;\times \;100\;\left(\%\right)$$
where PCS is the protein content in the supernatant after centrifugation, and TPC is the total protein content present in the protein sample.

#### 3.8.2. Water-Binding Capacity and Oil-Absorption Capacity

Water-binding capacity (WBC) of native and acetylated PPC was determined according to the method described by Jeżowski et al. [71]. For this, 1 g of powdered sample was weighed in a test tube containing 20 mL of distilled water. It was allowed to mix in a laboratory mixer for 60 s and was allowed to stand for 15 min at ambient temperature. This slurry was centrifuged at 4500× *g* for 15 min (Rotofix 32A by Hettich, Tuttlingen, Germany). The separated solid was oven-dried. WBC was expressed as the amount of water (g) absorbed by 1 g of the preparation.

Oil-absorption capacity (OAC) was determined using the method of Wu et al. [72]. Briefly, 1 g of protein preparation was weighed in the test tube and mixed with 15 mL of rapeseed oil using a Vortex mixer. Samples were allowed to stand for 30 min. The resulting protein–oil mixture was separated using a centrifuge (4000 g; Rotofix 32A by Hettich, Tuttlingen, Germany) for 10 min. Immediately after centrifugation, the supernatant was carefully poured into a 15 mL graduated cylinder, and the volumes were recorded. OAC was expressed as the amount of oil (mL) absorbed by 1 g of the preparation.

#### 3.8.3. Effect of pH on Foaming Capacity and Stability

Foam capacity (FC) and stability (FS) were measured according to the method of Jeżowski et al. [71]. Briefly, one gram of the preparation was weighed into the tube and 200 mL of distilled water was added to it. The resulting mixture was adjusted to the appropriate pH (2, 4, 6, 8, 10 or 12) using 0.5 M NaOH or 0.5 HCl. The sample was then homogenized for 2 min at 16,000 rpm (T25 basic ULTRA-TURRAX^®^; IKA Werke, Staufen, Germany). The beaten sample was immediately transferred to a measuring cylinder, where the total foam volume was determined after 0, 5, 10, 30 and 60 min. FC and FS were calculated according to the following equations (Equations (2) and (3)):(2)$$\mathrm{FC}=\frac{\mathrm{VA}}{\mathrm{VB}}\;\times \;100\;\left(\%\right)$$
where VA denotes the volume after whipping (mL), and VB is the volume before whipping (mL).
(3)$$\mathrm{FS}=\frac{\mathrm{VB}}{\mathrm{VT}}\;\times \;100\;\left(\%\right)$$
where VB denotes the volume before whipping (cm^3^), and VT is the volume after a certain time (mL).

#### 3.8.4. Effect of pH on Emulsifying Properties

Emulsification activity (EA) and stability (ES) were estimated by the method of Miedzianka et al. [24] with slight modifications (Equations (4) and (5)). The protein suspensions with added oil were mixed using an Ultra-turrax T-18 model homogenizer (IKA-Werke GmbH and Co. KG, Staufen, Germany) to produce a crude emulsion for 1 min at 20,000 rpm. Then, they were centrifuged at 3000× *g* for 10 min. Emulsion stability was determined by centrifugation after heating at the temperature of 80 °C for 30 min.
(4)$$\mathrm{EA}\;\left(\%\right)=\frac{\mathrm{height}\;\mathrm{of}\;\mathrm{emulsified}\;\mathrm{layer}\;\mathrm{in}\;\mathrm{the}\;\mathrm{tube}}{\mathrm{height}\;\mathrm{of}\;\mathrm{the}\;\mathrm{total}\;\mathrm{contents}\;\mathrm{in}\;\mathrm{the}\;\mathrm{tube}}\;\times \;100\;$$
(5)$$\mathrm{ES}\;\left(\%\right)=\frac{\mathrm{height}\;\mathrm{of}\;\mathrm{emulsified}\;\mathrm{layer}\;\mathrm{after}\;\mathrm{heating}}{\mathrm{height}\;\mathrm{of}\;\mathrm{the}\;\mathrm{total}\;\mathrm{contents}\;\mathrm{before}\;\mathrm{heating}}\;\times \;100\;$$

### 3.9. Statistical Analysis

Statistical analysis of all data was performed using one-way analysis of variance (ANOVA). The analysis of chemical composition and functional properties were performed in duplicate and in triplicate, respectively. Duncan’s range test was used to determine the differences among samples with a probability level of 0.05. Statistical analysis and standard deviations were determined using Statistica v. 13.3 software (StatSoft Inc., Tulsa, OK, USA).

## 4. Conclusions

Protein acylation with acetic acid anhydride used in this study is one of the methods of shaping the functional properties of proteins. However, despite the large number of publications dealing with the problem of improving the functional properties of protein preparations, it is still difficult to find universal regularities in the explanation of the mechanisms of the relationship between the structure of modified proteins and their function. Acetylation generally improves the functional properties of the obtained commercial pumpkin protein preparations, mainly protein solubility, as well as water-binding and oil-absorption capacity, which extends the possibilities of their use in the food industry.

## Figures and Tables

**Figure 1 molecules-26-01575-f001:**
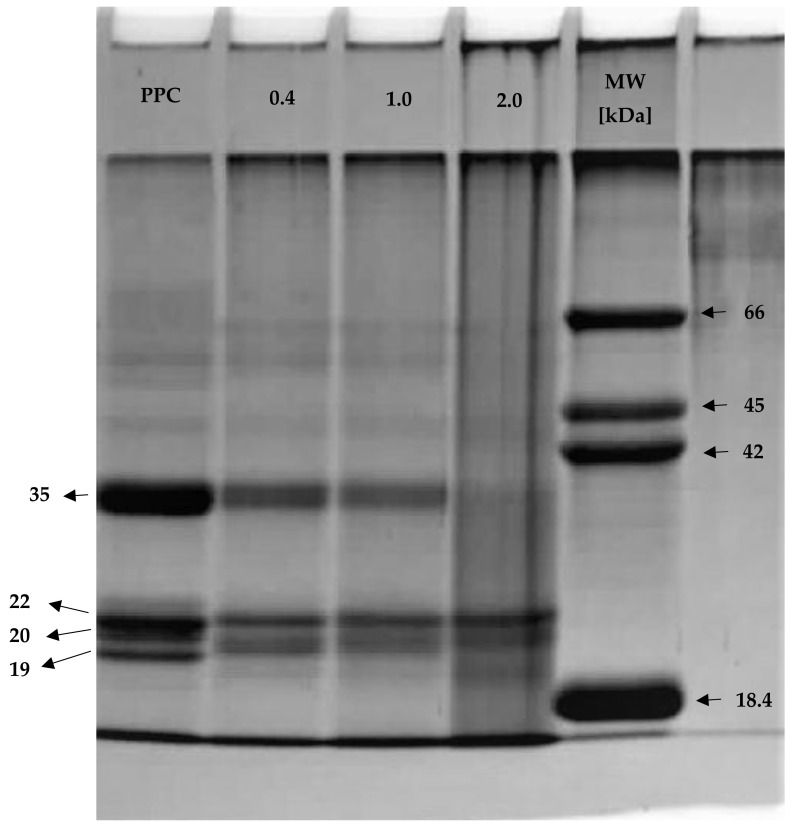
SDS-PAGE of native and acetylated PPC. PPC—pumpkin protein concentrate; 0.4, 1.0, 2.0—pumpkin protein preparations after acetylation conducted with different concentrations of acetic anhydride (mL/g); MW—molecular weight marker prepared at laboratory (18.4 kDa (β-lactoglobulin), 42, 45 kDa (ovalbumin), 66 kDa (bovine serum albumin).

**Figure 2 molecules-26-01575-f002:**
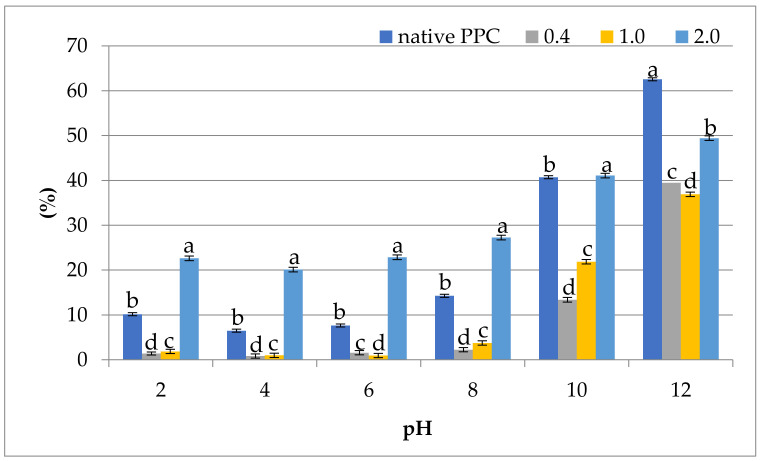
Effect of anhydride-to-protein ratio on protein solubility (PS) at different pH of native and acetylated PPC. Values are means ± standard deviation; ^a,b,c,d^—the same letters within the same pH were not significantly different; PPC—pumpkin protein concentrate; 0.4, 1.0, 2.0—pumpkin protein preparations after acetylation conducted with different concentrations of acetic anhydride (mL/g).

**Figure 3 molecules-26-01575-f003:**
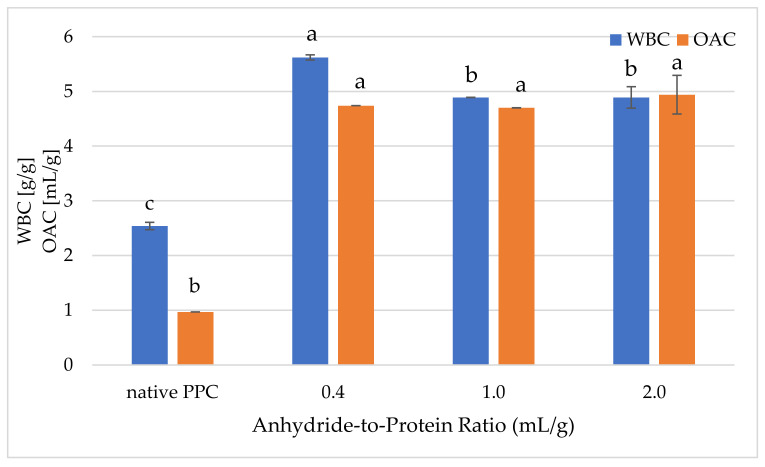
Effect of anhydride-to-protein ratio on water-binding capacity (WBC) and oil-absorption capacity (OAC) of native and acetylated PPC. ^a,b,c^—the same letters within the same analysis were not significantly different; PPC—pumpkin protein concentrate; 0.4, 1.0, 2.0—pumpkin protein preparations after acetylation conducted with different concentrations of acetic anhydride (mL/g).

**Figure 4 molecules-26-01575-f004:**
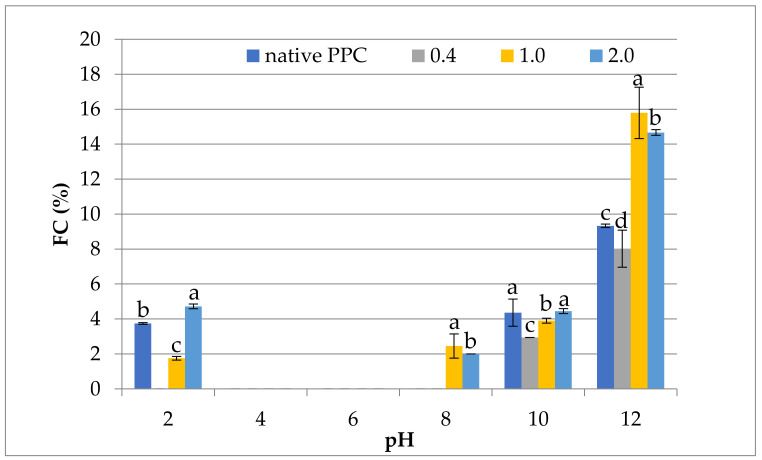
Effect of anhydride-to-protein ratio on foam capacity (FC) at different pH of native and acetylated PPC. ^a,b,c^—the same letters within the same pH were not significantly different; PPC—pumpkin protein concentrate; 0.4, 1.0, 2.0—pumpkin protein preparations after acetylation conducted with different concentrations of acetic anhydride (mL/g).

**Figure 5 molecules-26-01575-f005:**
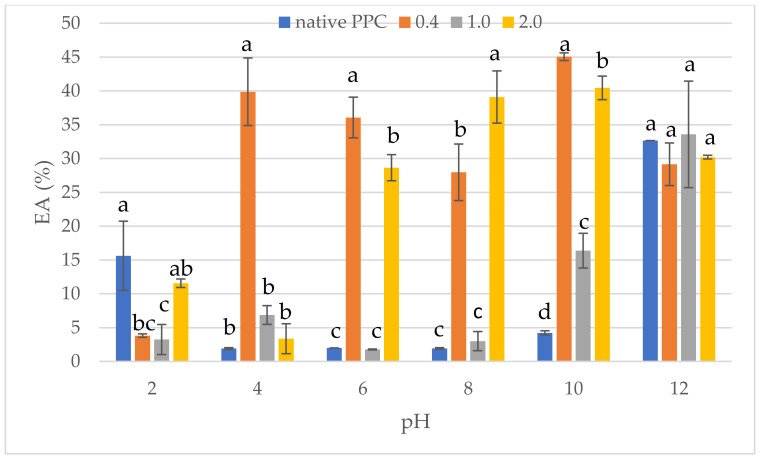
Effect of anhydride-to-protein ratio on emulsion activity (EA) at a different pH of native and acetylated PPC. ^a,b,c,d^—the same letters within the same pH were not significantly different; PPC—pumpkin protein concentrate; 0.4, 1.0, 2.0—pumpkin protein preparations after acetylation conducted with different concentrations of acetic anhydride (mL/g).

**Figure 6 molecules-26-01575-f006:**
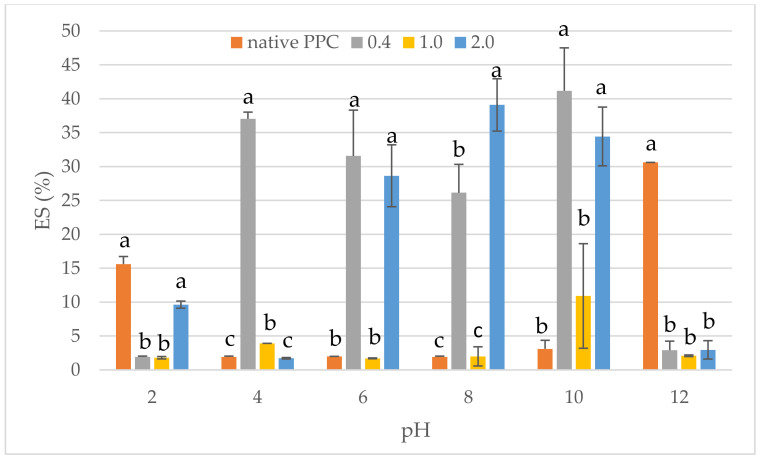
Effect of anhydride-to-protein ratio on emulsion activity (ES) at different pH of native and acetylated PPC. ^a,b,c^—the same letters within the same pH were not significantly different; PPC—pumpkin protein concentrate; 0.4, 1.0, 2.0—pumpkin protein preparations after acetylation conducted with different concentrations of acetic anhydride (mL/g).

**Table 1 molecules-26-01575-t001:** Characteristics of native and acetylated PPC.

Anhydride-to-Protein Ratio (mL/g)	Dry Matter	Protein Content	Fat	Ash
	(g/100g)			
Native PPC	97.49 ± 0.39 ^b^	65.11 ± 0.76 ^a^	9.17 ± 0.17 ^a^	8.36 ± 0.01 ^b^
0.4	94.90 ± 0.01 ^c^	63.24 ± 2.68 ^a^	8.91 ± 0.15 ^b^	8.68 ± 0.02 ^a^
1.0	99.01 ± 0.24 ^a^	53.52 ± 0.88 ^b^	8.88 ± 0.13 ^b^	7.99 ± 0.02 ^c^
2.0	98.80 ± 0.01 ^a^	62.24 ± 2.99 ^a^	8.89 ± 0.09 ^b^	5.49 ± 0.01 ^d^

Values are means ± standard deviation; *n* = 2; ^a,b,c,d^—the same letters within the same row were not significantly different; PPC—pumpkin protein concentrate; 0.4, 1.0, 2.0—pumpkin protein preparations after acetylation conducted with different concentrations of acetic anhydride.

**Table 2 molecules-26-01575-t002:** Amino acid concentration (g/100 g protein) in native and acetylated PPC.

Amino Acids	Native PPC	Anhydride-to-Protein Ratio (mL/g)		
		0.4	1.0	2.0
IAA *				
Leucine	6.67 ± 2.43 ^a^	6.56 ± 1.01 ^b^	6.13 ± 1.00 ^c^	6.60 ± 0.72 ^ab^
Isoleucine	4.15 ± 0.34 ^a^	3.80 ± 1.07 ^b^	3.55 ± 0.32 ^c^	3.88 ± 0.42 ^b^
Methionine	1.62 ± 0.51 ^a^	1.27 ± 1.21 ^b^	0.97 ± 0.48 ^d^	1.16 ± 0.85 ^c^
Cysteine	0.42 ± 0.88 ^a^	0.32 ± 0.44 ^b^	0.23 ± 0.01 ^d^	0.27 ± 0.25 ^c^
Phenyloalanine	4.51 ± 2.26 ^a^	3.40 ± 1.56 ^b^	2.95 ± 0.18 ^d^	3.05 ± 0.87 ^c^
Threonine	1.99 ± 0.07 ^a^	1.97 ± 2.19 ^b^	0.84 ± 0.34 ^d^	1.14 ± 0.26 ^c^
Lysine	3.32 ± 0.34 ^a^	2.69 ± 1.76 ^b^	2.02 ± 0.86 ^c^	1.89 ± 0.69 ^d^
Tyrosine	1.70 ± 0.26 ^a^	1.21 ± 0.45 ^b^	0.75 ± 0.51 ^d^	1.13 ± 0.30 ^c^
Valine	4.10 ± 0.86 ^a^	3.15 ± 0.86 ^b^	2.72 ± 0.30 ^c^	2.56 ± 0.52 ^c^
DAA **				
Aspartic acid	7.04 ± 0.72 ^a^	6.15 ± 0.72 ^d^	6.29 ± 0.79 ^b^	6.18 ± 0.95 ^c^
Glutamic acid	21.03 ± 4.26 ^a^	19.28 ± 2.81 ^c^	19.78 ± 0.93 ^b^	18.05 ± 1.58 ^d^
Serine	3.86 ± 1.07 ^a^	3.81 ± 1.27 ^b^	3.56 ± 0.23 ^d^	3.68 ± 1.92 ^c^
Glycine	2.83 ± 2.14 ^a^	2.44 ± 0.72 ^b^	1.73 ± 0.52 ^d^	2.07 ± 1.92 ^c^
Alanine	3.64 ± 0.57 ^d^	4.07 ± 0.57 ^a^	3.89 ± 0.38 ^b^	3.84 ± 0.33 ^c^
Histidine	1.72 ± 1.31 ^a^	1.61 ± 1.31 ^b^	1.49 ± 0.21 ^c^	1.60 ± 0.19 ^b^
Arginine	16.69 ± 0.51 ^a^	16.08 ± 0.91 ^c^	16.30 ± 2.83 ^b^	15.90 ± 0.17 ^d^
Proline	3.46 ± 1.90 ^b^	3.55 ± 2.24 ^a^	3.21 ± 0.97 ^d^	3.35 ± 1.05 ^c^
Total amino acids	88.63 ± 1.20 ^a^	85.98 ± 1.24 ^a^	81.07 ± 1.26 ^b^	79.88 ± 0.82 ^b^

Values are means ± standard deviation; *n* = 2; ^a,b,c,d^—the same letter in verse mean homogenous group; IAA *—indispensable amino acids; DAA **—dispensable amino acids.

**Table 3 molecules-26-01575-t003:** Degree of N-acetylation and in vitro digestibility of native and acetylated PPC.

Dose of Acetic Anhydride	Degree of N-Acylation	Amount of Protein Released into the Intestinal Fluid in Two-Step Digestion
(mL/g)	(%)	(%)
Native PPC	-	23.27 ± 0.01 ^b^
0.4	54.89 ± 0.20 ^c^	21.04 ± 0.01 ^c^
1.0	62.77 ± 0.33 ^b^	25.31 ± 0.01 ^a^
2.0	78.67 ± 0.12 ^a^	23.09 ± 0.01 ^b^

Values are means ± standard deviation; ^a,b,c^—the same letters within the same row were not significantly different; PPC—pumpkin protein concentrate; 0.4, 1.0, 2.0—pumpkin protein preparations after acetylation conducted with different concentrations of acetic anhydride.

**Table 4 molecules-26-01575-t004:** Results of the foam stability (FS, %) at different pH of native and acetylated PPC.

Foam Stability (%)								
Type of Preparation		Time	pH					
			2	4	6	8	10	12
**Native PPC**		5 min	1.9	0	0	0	4.5	8.0
		10 min	1.9	0	0	0	4.5	3.7
		30 min	1.4	0	0	0	3.0	1.9
		60 min	1.4	0	0	0	2.0	1.9
**Anhydride-to-Protein Ratio (mL/g)**	**0.4**	5 min	0	0	0	0	0	5.8
		10 min	0	0	0	0	0	3.6
		30 min	0	0	0	0	0	2.2
		60 min	0	0	0	0	0	1.7
	**1.0**	5 min	0	0	0	0	0	4.3
		10 min	0	0	0	0	0	3.5
		30 min	0	0	0	0	0	3.5
		60 min	0	0	0	0	0	3.5
	**2.0**	5 min	0	0	0	0	0	8.7
		10 min	0	0	0	0	0	8.6
		30 min	0	0	0	0	0	0
		60 min	0	0	0	0	0	0

PPC—pumpkin protein concentrate; 0.4, 1.0, 2.0—pumpkin protein preparations after acetylation conducted with different concentrations of acetic anhydride.

## Data Availability

Data is contained within the article.

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
