# Peer review of "Effect of Acetylation on Physicochemical and Functional Properties of Commercial Pumpkin Protein Concentrate"

_molecules, 2021, doi:10.3390/molecules26061575_

Round 1

Reviewer 1 Report

The paper is of good scientific quality, it provide an advance in the current knowledge on the industrial utilization of commercial pumpkin protein concentrate. The paper is interesting for researchers in food technology and food industry.

It is clear, only table 4 could be better presented.

Author Response

The authors are very grateful for valuable remarks, which have been taken into account in the revised version.

General: The paper has been corrected according to the Reviewer’s notes.

Answers for questions:

Comment 1: It is clear, only table 4 could be better presented.

Author response: The table 4 has been corrected according to the Reviewer’s suggestion.

Reviewer 2 Report

The paper “Effect of Acetylation on Physicochemical and Functional Properties of Commercial Pumpkin Protein Concentrate” was focused on analyzing the influence of pumpkin protein concentrate (PPC) acetylation on different induced changes of the functional properties such as: Protein solubility index, Water-binding capacity, and oil-absorption capacity, Foaming properties, Emulsifying properties. For this goal acetic anhydride was added at three different concentrations: 0.4, 1.0, and 2.0 mL/ 1 g of protein.

The article has relevant information and these investigations are important for the use of PPC as a food additive since it is known that their poor functional properties could limit its utilization.

The present paper is well structured, the presented results seem correct as well as the used experimental methodology and the discussion of the results and the references are relevant.

The Results and Discussion section, also presents the influence of acetylation on the chemical composition of the PPC and on the digestibility (in vitro) by comparing native and acetylated PPC. These results are also important for the research topic. I am wondering if a section regarding the toxicological effect is not necessary, at least by citing other relevant studies regarding the safety evaluation of the modified proteins.

Author Response

The authors are very grateful for valuable remarks, which have been taken into account in the revised version.

General: The paper has been corrected according to the Reviewer’s notes.

Answers for questions:

Comment 1: I am wondering if a section regarding the toxicological effect is not necessary, at least by citing other relevant studies regarding the safety evaluation of the modified proteins.

Author response: The actual issue is very important; therefore, an explanation has been added to the introduction: This chemical modification is considered safe because acetic anhydride is rapidly hydrolyzed (half-life 4.4 min.) to acetic acid which is readily biodegradable. In the atmosphere, it is converted to acetic acid which is subject to photooxidative degradation (half-life 22 days). Toxicity to aquatic organisms is moderate (18 to 3400 mg/l), but it persists only for a short time due to its rapid hydrolysis to acetate/acetic acid. Acetic acid is further used as a food and animal feed additive, or a preservative in pickles (lines 69-74).

Reviewer 3 Report

The study of Miedzianka examines the effect of acetylation of pumpkin protein concentrates on its physicochemical and functional properties. Three different ratios of acetic anhydride-to-protein were tested in order to assess whether the effect was dose depended.  In general, the study is well structured and various parameters, both physicochemical and functional were examined.  I have some concerns about the appropriateness of some methodologies and also about the results of the statistical analysis for some parameters (which was based on two experimental repetitions only) because it seems that some values ​​do not support the declared result. Also, no final correlation of the results with the effect of the dose has been made, which should probably be further discussed. Moreover, I believe that there should be a more extensive discussion on the correlates the overall value of acetylation with the improvement of the functional properties of the final product in relation to the proposed applications-uses.

Specific comments

Line 57: acetylation instead of acylation?

Lines 62-63: What is the range of acetic anhydride proposed in the literature for the acetylation of proteins?

Lines 74-76: Which are the targeted properties that are expected to be affected by acetylation?

Table 1: Please provide SD and statistical differences in the carbohydrate content

Lines 100-104: It is argued that the decrease in the protein content is due to probable hydrolysis. Shouldn’t this effect be dose depended? Where do you attribute the observation that proteins were degraded only for 1ml/g anhydride-to-protein ratio?   

 Lines 125-126: I am not sure that this argument is precise. There are actually seems to be a statistically different decreasing trend on the ash content for higher anhydride-to-protein ratios.

 Lines 128-129: I am not sure that this argument is totally correct. The study of Jandal et al (Ref.27) deals with milk whey proteins which have a high content of calcium salts, the transferring of which into the ionic form may increase the ash percentage of the concentrates. The effect is also reported to be affected by the salinity of the whey. Please revise

Lines 128-129: The meaning is not clear. Please revise. How does acetylation affect carbohydrates content based on the literature?

Table 2: The statistical analysis should be checked again.  

Lines 393:  Please provide a reference for the calculation of proteins from nitrogen

Lines 399-400: The indirect calculation of carbohydrates is not acceptable. Carbohydrates should be estimated using some standard analytical method  

Lines 410-420: I wonder if the methodology used for the estimation of the degree of N-acylation is appropriate. The estimation was based on the correlation of spectrophotometrically estimated free amino groups, assuming that the control sample of non-acetylated protein concentrate corresponds to 100%. However, it is mentioned that the acetylation process causes hydrolysis of proteins, and as such increase of free amino grours. Was that taken somehow into account?

Author Response

The authors are very grateful for valuable remarks, which have been taken into account in the revised version.

General: The paper has been corrected according to the Reviewer’s notes.

Answers for questions:

General comment: I have some concerns about the appropriateness of some methodologies and also about the results of the statistical analysis for some parameters (which was based on two experimental repetitions only) because it seems that some values ​​do not support the declared result. Also, no final correlation of the results with the effect of the dose has been made, which should probably be further discussed. Moreover, I believe that there should be a more extensive discussion on the correlates the overall value of acetylation with the improvement of the functional properties of the final product in relation to the proposed applications-uses.

Author response to general comment: We fully agree that repeating the experiment twice may raise doubts. However, the influence of acetylation on the properties of various proteins is our leading area of ​​research. The acetylation effects of pumpkin proteins are very similar to those obtained for example for potato proteins (Miedzianka et al., 2012; ref. 24). This is a confirmation of the reliability of the research results.

In fact, some of the results were discussed inaccurately, therefore we made a number of corrections:

- The sentence:’The chemical composition of native and acetylated PPC is presented in Table 1. All analysed samples differed in protein, fat and ash content.’ has been changed on: ‘The chemical composition of native and acetylated PPC is presented in Table 1. All analysed samples differed in dry matter, fat, and ash content’ (lines 87-88)

- The sentence has been added: ‘In most cases, acetylation had no effect on the protein content. No statistically significant differences were found between the control and the most experimental samples’ (lines 97-99).

- To the statement: ‘Acetylation reduced the fat content in modified PPC. As shown in Table 1, the fat content ranged from an average of 9.17 g/100 g before chemical modification to an average of 8.89 g/100 g in modified samples. However, the acetylated samples were not significantly differed in fat content’ the sentence: ‘Increasing the amount of anhydride did not affect the level of fat’ has been added (lines 119-120).

- To the 2.5.4. Emulsifying properties section the sentence has been added: ‘Furthermore El- Adawy (ref. 21) proved that the best effect for mung bean proteins is obtained using 0.4-0.6 g/g of acetic anhydride. This proves that apart from the conditions of acetylation, the origin (properties) of proteins influences the change of their functional properties’ (lines 357-360).

Comment 1: Line 57: acetylation instead of acylation?

Author response: The phrase ‘acylation’ has been removed and replaced with ‘acetylation’.

Comment 2: Lines 62-63: What is the range of acetic anhydride proposed in the literature for the acetylation of proteins?

Author response: The range of acetic anhydride proposed in the literature for acetylation of proteins is 0.2 – 1.0 ml/ g protein. This information has been added into line 62.

Comment 3: Lines 74-76: Which are the targeted properties that are expected to be affected by acetylation?

Author response: The assumption was the selection of acetylation conditions affecting the improvement of water binding, oil absorption, solubility, foaming and emulsifying properties while limiting the negative impact on changes in the chemical composition and digestibility of proteins (lines 77-79).

Comment 4: Table 1: Please provide SD and statistical differences in the carbohydrate content

Author response: Content of Table 1 has been modified also in response to the linked comments of the Reviewer. The data concerning the carbohydrate content has been removed from the article after careful consideration based on the scientific literature studies where authors when analyzing the effect of chemical modification of plant-derived protein preparations did not focus on carbohydrate content. 

Comment 5: Lines 100-104: It is argued that the decrease in the protein content is due to probable hydrolysis. Shouldn’t this effect be dose depended? Where do you attribute the observation that proteins were degraded only for 1ml/g anhydride-to-protein ratio?

Author response: We are very ashamed of this not fully thought-out statement. It is well known that the protein level determined by the Kjeldahl method does not change after hydrolysis because nitrogen remains unchanged. However, many authors point to a reduction or increase in protein levels after acetylation (El-Adawy, ref. 21; Lawal and Adebowale, ref. 22; Lawal et al., ref. 23 and Miedzianka et al., ref. 24). In the literature on the subject, this phenomenon has not been mentioned.

The protein determination in the subject trials was repeated. Exactly the same results were obtained. We cannot explain what effect the dose of 1.0 mL/g has on the protein level. However, the same relationship was observed during the acetylation of potato proteins (ref. 24). In conclusion, we decided to remove the statement (‘This reduced content of protein in acetylated samples could result from the partial hydrolysis of proteins in the direct place of contact with the acetic anhydride and the rinsing process after modification’) from the discussion part.

Comment 6: Lines 125-126: I am not sure that this argument is precise. There are actually seems to be a statistically different decreasing trend on the ash content for higher anhydride-to-protein ratios.

Author response: I agree. There are statistically different decreasing trend on the ash content for higher anhydride-to-protein ratios. The sentence: ‘Furthermore, there was no trend observed between the dose of acetic anhydride and the ash content of analysed samples’ has been removed from the manuscript.

Comment 7: Lines 128-129: I am not sure that this argument is totally correct. The study of Jandal et al (Ref.27) deals with milk whey proteins which have a high content of calcium salts, the transferring of which into the ionic form may increase the ash percentage of the concentrates. The effect is also reported to be affected by the salinity of the whey. Please revise.

Author response: I agree. This reference did not fit well. It has been removed from this part and new sentences have been added: ‘Similar finding were observed by Lawal and Adebowale (22), who acetylated the jack bean protein. This decreasing ash content in acetylated protein preparations can be associated with more frequent removal of excess modifying reagent (samples modified in the amount of 2.0 mL / g of protein were washed 5 times)’ (lines 135-138).

Comment 8: Lines 128-129: The meaning is not clear. Please revise. How does acetylation affect carbohydrates content based on the literature?

Author response: We fully agree that the comment is unclear, has no bearing on acetylation and should be redrafted. However, after careful consideration, we decided to remove the carbohydrate analysis from the article. When undertaking an article on the properties of protein preparations derived from pumpkin seeds, our aim was mainly to check the chosen functional properties of these proteins. Moreover, in the available world literature there is no information on the effect of acetylation on the carbohydrate content in protein preparations of plant origin.

Comment 9: Table 2: The statistical analysis should be checked again.

Author response: The statistical analysis in Table 2 was re-checked.

Comment 10: Lines 393:  Please provide a reference for the calculation of proteins from nitrogen.

Author response: The reference for the calculation of proteins from nitrogen has been provided: Mariotti et al. (2008) (ref. 63).

Comment 11: Lines 399-400: The indirect calculation of carbohydrates is not acceptable. Carbohydrates should be estimated using some standard analytical method.

Author response: We agree that the carbohydrates should be estimated using some standard analytical method. However, in almost all studies concerning the acetylation of plant-derived protein preparations (i.e. Bora, ref. 53; El-Adawy, ref. 21), there is no information regarding the influence of this chemical modification on the content of carbohydrates. Therefore, we decided to remove this analysis from our manuscript.

Comment 12: Lines 410-420: I wonder if the methodology used for the estimation of the degree of N-acylation is appropriate. The estimation was based on the correlation of spectrophotometrically estimated free amino groups, assuming that the control sample of non-acetylated protein concentrate corresponds to 100%. However, it is mentioned that the acetylation process causes hydrolysis of proteins, and as such increase of free amino groups. Was that taken somehow into account?

Author response: The applied methodology used for the estimation of the degree of N-acylation is the most popular one. It is used by Lawal and Adebowale (ref. 22) or Zhao et al.  (ref. 16). This method was proposed among others for protein modification studies. Information that acetylation caused the hydrolysis of proteins was removed from the manuscript. We are very ashamed of this not fully thought-out statement.

Reviewer 4 Report

Manuscript "Effect of Acetylation on Physicochemical and Functional Prop-2erties of Commercial Pumpkin Protein Concentrate" confirms the well-known facts about plant proteins and adds nothing new to this problem. The reviewer proposes publication in a journal of lower scientific quality. 

Author Response

Author response: The problem of chemical modification of vegetable proteins, also considered recently in terms of the use of the obtained preparations for the production of food, is aimed at examining the influence of various modifications on the shaping of physical and functional properties of the modifications and determining the relationship between the degree of modification and these properties of preparations. Authors who deal with the problem of pumpkin protein modification focus primarily on studying the course and effects of enzymatic protein hydrolysis in terms of obtaining products with biological activity. There is no information in the available literature on research on the chemical modification of pumpkin proteins.

Round 2

Reviewer 3 Report

The comments were adequately addressed and the changes made in the MS are sufficient.

Reviewer 4 Report

In reviewer opinion manuscript is ready to publication.